# Nutraceutical Profile Characterization in Apricot (*Prunus armeniaca* L.) Fruits

**DOI:** 10.3390/plants14071000

**Published:** 2025-03-22

**Authors:** Germán Ortuño-Hernández, Marta Silva, Rosa Toledo, Helena Ramos, Ana Reis-Mendes, David Ruiz, Pedro Martínez-Gómez, Isabel M. P. L. V. O. Ferreira, Juan Alfonso Salazar

**Affiliations:** 1Fruit Breeding Group, Department of Plant Breeding, CEBAS-CSIC (Centro de Edafología y Biología Aplicada del Segura-Consejo Superior de Investigaciones Científicas), Campus Universitario Espinardo, E-30100 Murcia, Spain; gortuno@cebas.csic.es (G.O.-H.); druiz@cebas.csic.es (D.R.); pmartinez@cebas.csic.es (P.M.-G.); 2LAQV/REQUIMTE, Departamento de Ciências Químicas, Laboratório de Bromatologia e Hidrologia, Faculdade de Farmácia, Universidade do Porto, Rua de Jorge Viterbo Ferreira n°. 228, 4050-313 Porto, Portugal; marta.sdds@gmail.com (M.S.); ahelenaramos@gmail.com (H.R.); afreis.mendes@gmail.com (A.R.-M.); isabel.ferreira@ff.up.pt (I.M.P.L.V.O.F.); 3Metabolomics Platform of CEBAS-CSIC (Centro de Edafología y Biología Aplicada del Segura-Consejo Superior de Investigaciones Científicas), Campus Universitario Espinardo, E-30100 Murcia, Spain; rtoledo@cebas.csic.es

**Keywords:** stone fruits, functional foods, health benefits, nutribreeding, primary metabolites, secondary metabolites, plant breeding

## Abstract

This study characterizes the metabolomic profiles of three reference apricot cultivars (‘Bergeron’, ‘Currot’, and ‘Goldrich’) using ^1^H NMR spectroscopy and untargeted UPLC-QToF MS/MS to support plant breeding by correlating metabolomic data with fruit phenotyping. The primary objective was to identify and quantify the key metabolites influencing fruit quality from a nutraceutical perspective. The analysis revealed significant differences in primary and secondary metabolites among the cultivars. ‘Bergeron’ and ‘Goldrich’ exhibited higher concentrations of organic acids (109 mg/g malate in ‘Bergeron’ and 202 mg/g citrate in ‘Goldrich’), flavonoids such as epicatechin (0.44 mg/g and 0.79 mg/g, respectively), and sucrose (464 mg/g and 546 mg/g), contributing to their acidity-to-sugar balance. Conversely, ‘Currot’ showed higher levels of amino acids (24.44 mg/g asparagine) and sugars, particularly fructose and glucose (79 mg/g and 180 mg/g), enhancing its characteristic sweetness. These findings suggest that metabolomic profiling can provide valuable insights into the biochemical pathways underlying apricot quality traits, aiding in the selection of cultivars with desirable characteristics. The integration of phenotyping data with ^1^H NMR and UPLC-QToF MS/MS offers a comprehensive approach to understanding apricot metabolomic diversity, crucial for breeding high-quality, nutritionally enriched fruits that meet market demands.

## 1. Introduction

The characterization of the metabolomic profile of fruit plant cultivars is becoming an increasingly important tool in plant breeding. It provides a deep understanding of the metabolites that influence fruit quality [1].

Apricot (*Prunus armeniaca* L.), originating from Central Asia, is widely cultivated in temperate climates around the world. Apricot fruit exhibits a wide range of variability in size, shape, and color, depending on the cultivar [2]. This fruit is highly valued for its high content of essential nutrients, dietary fiber, and antioxidants [3]. Among its primary metabolites, apricot fruit is a rich source of sucrose, glucose, and fructose, which contribute to its sweet taste [4]. Additionally, they contain significant levels of amino acids, with alanine, aspartic acid, and glutamine being the most prevalent. Organic acids, such as citric and malic acid, further contribute to the fruit’s distinctive tartness [5]. In terms of secondary metabolites, apricots are a notable source of various phenolic acids and flavonoids, compounds known for their potent antioxidant activity [6]. Phenolic acids, such as gallic acid and chlorogenic acid, and flavonoids, including flavonols like quercetin and anthocyanins, are particularly abundant [7]. Furthermore, glycosylated forms of these flavonoids, such as quercetin glycosides, contribute to the fruit’s bioactive profile [8]. Together, these metabolites work synergistically to enhance overall health and improve the organoleptic qualities of apricots, making this fruit not only nutritious, but also functionally beneficial [9].

The versatility of the apricot is reflected in its application across a wide range of products, from fresh consumption to processing into jams, juices, and dried fruits, highlighting its importance in both human diets and the food industry [10]. According to data from FAOSTAT 2023 [11], the demand for apricots, both fresh and processed, is steadily increasing. The data reveal an increase in fresh apricot production by 16.9% from 2010 to 2022, and a notable 39.35% rise in processed apricot production from 2009/2010 to 2022/2023.

This sustained growth in apricot production and demand underscores the need for genetic and agronomic improvement programs. Such programs are essential for increasing productivity, enhancing fruit quality, and ensuring the sustainability of the crop in the face of climatic and phytosanitary challenges. Apart from CEBAS CSIC in Spain, known for developing new apricot cultivars with improved traits, notable apricot breeding programs include RosBREED in the USA [12], which focuses on enhancing disease resistance and fruit quality, as well as prominent initiatives in China, Italy, and Turkey [13,14,15], which have extensive programs aimed at increasing yield and fruit quality. However, the metabolomic diversity among different apricot cultivars has not yet been fully explored, representing a significant opportunity for the improvement of these varieties.

Since the 1990s, the research team at CEBAS-CSIC has focused on apricot plant breeding, generating new cultivars that cover a wide range of harvest times and fruit qualities. In this context, some of the contrasting cultivars used as parents in segregating populations included ‘Bergeron’, ‘Currot’, and ‘Goldrich’, selected for their relevance to plant breeding programs due to their distinct characteristics [16]. These cultivars were used as parental lines, with ‘Currot’ as a common parent, resulting in the populations ‘Bergeron’ × ‘Currot’ and ‘Goldrich’ × ‘Currot’, in which numerous QTLs (Quantitative Trait Loci) related to phenology and fruit quality traits have been identified [17]. Additionally, post-harvest trials have been conducted to evaluate and enhance the fruit’s shelf life [18,19].

There are very few studies on the quantification of metabolites in apricots using ^1^H Nuclear Magnetic Resonance (NMR), and even fewer that combine this approach with untargeted Ultra-Performance Liquid Chromatography coupled with Quadrupole Time-of-Flight mass spectrometry/mass spectrometry (UPLC QToF MS/MS) for polyphenolic compounds with a particular emphasis on phenolic acids, flavonoids, and glycosides and glucosylated compounds. Nonetheless, ^1^H NMR has proven to be a highly effective tool for identifying chemical compounds and establishing correlations between the chemical composition and physical properties of different cultivars. The research effectively demonstrated this by successfully identifying compounds including epicatechin, myo-inositol, and sucrose, among others [1]. Additionally, untargeted UPLC QToF MS/MS is an advanced analytical technique used for the separation, identification, and quantification of a wide range of chemical compounds in a sample [20]. Its applicability makes it ideal for exploration studies and the discovery of new compounds and potential biomarkers [21].

When untargeted UPLC QToF MS/MS is combined with Nuclear Magnetic Resonance (NMR) data and phenotyping, a comprehensive metabolomic profile can be constructed. This study aims to characterize and compare the metabolomic profiles of three apricot cultivars of great interest in the CEBAS-CSIC breeding program, relating them to their pomological characteristics. Applying this methodology to assess variability among parental lines at the commercial maturity stage is crucial, as we aim to explore the possibility of studying these compounds in apricot progenies derived from these parents to identify marker–trait associations. Integrating phenotyping, NMR, and untargeted UPLC QToF MS/MS in the study of apricot metabolomics represents an innovative and comprehensive approach to plant improvement. This work will not only contribute to the scientific knowledge of apricot metabolomic diversity, but will also provide practical tools for plant breeders. The valuable information obtained will aid in selecting genotypes with desirable traits, facilitating the obtention of new apricot varieties with enhanced organoleptic and nutritional properties. This will help meet the growing demand for high-quality, nutritious fruits in the global market.

## 2. Results

### 2.1. Evaluation of Pomological Traits Through Phenotyping

Pomological traits were analyzed in three apricot cultivars—‘Bergeron’, ‘Goldrich’, and ‘Currot’—including the ripening date, fruit weight, the index of absorbance difference (I_AD_), skin color, blush color, percentage of blush color, flesh color, firmness, soluble solid content (SSC), and acidity (Table 1). The characterization of these traits sheds light on the differences among these cultivars, making them ideal for generating segregated populations and for cultivation depending on our purposes.

In terms of the ripening date, ‘Bergeron’ had the latest ripening, at 167 Julian days (16 June 2023), which could be beneficial for extending the harvest season. ‘Goldrich’ ripened at 157 Julian days (6 June 2023), while ‘Currot’ was the earliest, ripening in just 130 Julian days (10 May 2023). When it comes to fruit weight, ‘Goldrich’ stood out by producing the heaviest fruits, with an average weight of 86.57 g, significantly higher than ‘Bergeron’ (40.41 g) and ‘Currot’ (38.84 g), making it potentially more appealing to consumers who prefer larger fruits.

Regarding appearance, ‘Currot’ exhibited the highest values for skin color and flesh color (h° ≈ 100), indicating a more yellow appearance. This trait could be advantageous in the market, where consumers tend to prefer visually lighter yellow fruits. ‘Bergeron’ and ‘Goldrich’ had very similar values, both displaying orange coloration. However, ‘Bergeron’ stood out with a higher percentage of blush (26%), with an intense red color, giving it a distinctive hue (h° ≈ 48). Although ‘Currot’ also had a slightly lower percentage of blush (18%), its red blush color was less intense (h° ≈ 60). Considering the change in fruit color as the harvest criterion, ‘Bergeron’ was harvested with the highest I_AD_ (0.34), despite being the cultivar with the most orange coloration and the highest red blush compared to the other two cultivars. With respect to firmness, ‘Currot’ was the firmest cultivar (F ≈ 50 N). However, the other cultivars exhibited similar firmness values, suggesting a consistent pattern in the color/firmness relationship at the optimal harvest stage.

Finally, the soluble solid content (SSC), as indicator of sugar content and, therefore, sweetness, was the highest in ‘Bergeron’ (≈14°Brix), followed by ‘Goldrich’ (≈13°Brix) and ‘Currot’ (≈12°Brix), suggesting that ‘Bergeron’ might have been the sweetest of the three. Nevertheless, acidity is a crucial factor for the fruit’s flavor balance. ‘Goldrich’ was the most acidic (2.64 g malic acid/100 mL), while ‘Currot’ had the lowest acidity (1.44 g malic acid/100 mL). Considering these values, ‘Currot’ had a less pronounced acidic flavor, allowing the sweetness of the sugars to stand out more.

### 2.2. Evaluation of Metabolites Using Proton Nuclear Magnetic Resonance (^1^H NMR)

Using ^1^H NMR analytical techniques, we identified and quantified various compounds present in the three studied cultivars, ‘Bergeron’, ‘Currot’, and ‘Goldrich’. These compounds are categorized into the following classes: amino acids, alcohols, alkaloids, carbohydrates, flavonoids, organic acids, and polyphenol derivatives (Table 2). The results obtained for each class of compounds are presented below.

#### 2.2.1. Amino Acids

Six amino acids were identified in the samples. Among them, alanine (AA1) and asparagine (AA2) stood out due to their higher concentrations. The alanine concentration was significantly lower in ‘Bergeron’ (0.47 mg/g) compared to ‘Currot’ and ‘Goldrich’ (0.80 and 0.88 mg/g, respectively). Asparagine exhibited a notably higher concentration in ‘Currot’ (24.44 mg/g) compared to ‘Bergeron’ (8.93 mg/g) and ‘Goldrich’ (13.27 mg/g). For isoleucine (AA3), phenylalanine (AA4), threonine (AA5), and valine (AA6), the concentrations were similar among the varieties, ranging between 0.1 and 0.3 mg/g, though they also showed significant variations.

#### 2.2.2. Carbohydrates

Five carbohydrates were identified. Two of the quantified monosaccharides, fructose (C1) and glucose (C2), showed significantly higher concentrations in ‘Currot’ (approximately 80 and 180 mg/g, respectively) compared to ‘Bergeron’ and ‘Goldrich’. However, the disaccharide sucrose (C4) had the highest concentration in ‘Goldrich’ (546.98 mg/g), which was considerably different from the 370.64 mg/g found in the ‘Currot’ cultivar (the lowest among the three). Regarding the concentration of myo-inositol (C3) and xylose (C5), no significant differences were observed between the cultivars, following a similar pattern with values around 2.5 mg/g.

#### 2.2.3. Flavonoids and Alkaloids

Epicatechin (F1), known for its antioxidant properties and health benefits, was the only flavonoid identified. Its concentration was twice as high in ‘Goldrich’ (0.8 mg/g) compared to ‘Bergeron’, and significantly higher than that found in ‘Currot’ (0.03 mg/g). Regarding alkaloids, trigonelline (AK1) showed no significant differences among the cultivars, with concentrations of around 0.04 mg/g.

#### 2.2.4. Organic Acids

Seven organic acids were identified based on their conjugate bases. Citrate (OA1), malate (OA4), and quinate (OA5) were notable for having the highest concentrations in the cultivars. The highest concentration of organic acids was found in the ‘Goldrich’ cultivar, with a citrate content of 202.27 mg/g, significantly higher than that found in ‘Bergeron’ and ‘Currot’, which had approximately 70 mg/g. For malate (OA4), ‘Bergeron’ had the highest concentration (109.16 mg/g), followed by ‘Currot’ and ‘Goldrich’, with 71.34 and 57.56 mg/g, respectively. Quinic acid exhibited a uniform distribution across the three cultivars, with values ranging between 40 and 50 mg/g. On the other hand, fumarate (OA3), succinate (OA6), and tartrate (OA7) showed concentrations below 1.5 mg/g in all three cultivars, maintaining similar value ranges. Additionally, formate (OA2) was not detected in any of the samples, with the detection limit of the equipment being greater than 10 µM (specifically > 0.009 mg/g in this case).

#### 2.2.5. Polyphenol Derivatives and Alcohols

The only identified polyphenol derivative, chlorogenate (PD1), exhibited variations in concentrations, being higher in ‘Goldrich’ compared to ‘Bergeron’ and ‘Currot’, although these differences were not statistically significant. Regarding alcohols, choline (AC1) showed no significant differences among the varieties, displaying a uniform distribution across all of them, with concentrations ranging between 0.03 and 0.06 mg/g.

### 2.3. Evaluation of Metabolites Using Untargeted UPLC QToF MS/MS

The results of the tentative identification of twelve compounds present in the sample are outlined below, organized into the following three main groups: phenolic acids, flavonoids, and glycosides and glucosylated compounds (Table 3). Each compound was tentatively identified using advanced mass spectrometry techniques, comparing theoretical masses with measured masses and cross-referencing the data with well-known metabolic databases such as KEGG and HMDB.

#### 2.3.1. Phenolic Acids

The phenolic acids identified in the sample include caffeic acid (PA1), coumaric acid (PA2), and ferulic acid (PA3). These phenolic acids were identified based on their dehydration products with adducts [M−H_2_O+H]^+^ and are found within a retention time range of 9–11 min. The error values were below 2 ppm, accompanied by mSigma values of less than 10, resulting in a very high level of identification confidence for these peaks. The tentative identification was further validated by cross-referencing with the KEGG database.

#### 2.3.2. Flavonoids

The flavonoids identified in the sample include catechin (F2), myricitrin (F3), quercetin (F4), and rutin (F5). These flavonoids were identified based on their protonated adducts [M+H]^+^. Catechin (F2) presented a retention time of 10.43 min, while the other three flavonoids were found within a retention time range of 13.5–14 min. The errors were all within ±2.2 ppm. Two of them had mSigma values below 10, resulting in a high level of identification confidence, while the other two had mSigma values between 10 and 30, which are acceptable for identification. The tentative identification was further validated by cross-referencing with the HMDB database.

#### 2.3.3. Glycosides and Glucosylated Compounds

The glycosides and glucosylated compounds identified in the sample include kiwiionoside (G1), neryl arabinofuranosyl-glucoside (G2), vanilloyl glucose (G3), zizybeoside I (G4), and 3-hydroxy-beta-ionol 3-[glucosyl-(1->6)-glucoside] (G5). These compounds were identified based on their sodium adducts [M+Na]^+^ and protonated adducts [M+H]^+^. The retention time values for these compounds were less uniform, presenting a wide range from 9.97 to 16.37 min. The error values were all within ±2.4 ppm. Three of them had mSigma values below 10, indicating a high level of identification confidence, while the other two had mSigma values of between 10 and 30, which are acceptable for identification. The tentative identification was further validated by cross-referencing with the HMDB database.

### 2.4. Non-Quantitative Area Values in Untargeted UPLC QToF MS/MS Analysis

This section presents the area values obtained through untargeted UPLC QToF MS/MS analysis (Table 4). The table below shows the tentatively identified compounds in three different cultivars, ‘Bergeron’, ‘Currot’, and ‘Goldrich’. The compounds are grouped into the following three main classes: phenolic acids, flavonoids, and glycosides and glucosylated compounds. The relative area (ra) values concerning internal control allow for a comparative analysis among different cultivars.

Regarding phenolic acids, caffeic acid (PA1) showed greater abundance in ‘Bergeron’ and ‘Goldrich’ compared to ‘Currot’, indicating a significant varietal difference. Coumaric acid (PA2) was most abundant in ‘Currot’, followed by ‘Bergeron’, and least in ‘Goldrich’, suggesting that specific metabolic pathways are active in ‘Currot’. Ferulic acid (PA3) presented the highest levels in ‘Goldrich’, moderate levels in ‘Bergeron’, and the lowest levels in ‘Currot’.

Concerning flavonoids, catechin (F2) was significantly more abundant in ‘Bergeron’ and ‘Goldrich’, while ‘Currot’ showed much lower levels. Myricitrin (F3) was most abundant in ‘Bergeron’, with ‘Goldrich’ and ‘Currot’ showing lower levels. The levels of quercetin (F4) and rutin (F5) were highest in ‘Goldrich’, moderate in ‘Bergeron’, and lowest in ‘Currot’. These differences reflect differential accumulation of these flavonoids among the cultivars.

Lastly, with respect to glycosides and glucosylated compounds, kiwiionoside (G1) showed the highest abundance in ‘Bergeron’ compared to ‘Currot’ and ‘Goldrich’. Neryl arabinofuranosyl-glucoside (G2) was most abundant in ‘Currot’, with ‘Bergeron’ and ‘Goldrich’ showing lower levels, indicating a unique glycosylation pattern in ‘Currot’, Vanilloyl glucose (G3) was most abundant in ‘Bergeron’, with ‘Currot’ and ‘Goldrich’ having significantly lower levels. Notably, zizybeoside I (G4) and 3-hydroxy-beta-ionol 3-[glucosyl-(1->6)-glucoside] (G5) were not detected in ‘Bergeron’ but were present in ‘Currot’ and ‘Goldrich’, suggesting that specific glycosylation mechanisms are active in these cultivars.

### 2.5. Comparative Metabolomic Profiling

The integration of both analytical techniques, ^1^H NMR and untargeted UPLC QToF MS/MS, resulted in the identification of a broad range of primary and secondary metabolites in our cultivars, thus generating a comprehensive compound profile for each one. To understand the differences among the various metabolomic profiles, a principal component analysis (PCA) was performed to assess the significance of metabolites in the cultivars, as well as the influence of pomological traits (Figure 1).

The majority of the variation is explained by the *x*-axis (PC1), accounting for 53.1%, with this axis differentiating ‘Currot’ from ‘Bergeron’ and ‘Goldrich’. In the PCA, it was observed that the metabolites contributing most significantly to this variation in ‘Currot’ were primarily amino acids (AA2 and AA5), carbohydrates (C1 and C2), and glycosides and glucosylated compounds (G2 and G5), as well as the high fruit color values, reflecting a more yellow and less red hue in the case of the blush. Conversely, the most relevant metabolites in the opposite direction of the axis, representing ‘Bergeron’ and ‘Goldrich’, were organic acids and phenolic acids (OA1, OA3, OA4, OA5, OA6, PA1, and PA3), all identified flavonoids (F1, F2, F3, F4, and F5), and the carbohydrate sucrose (C4). The most prominent pomological traits included the ripening date, I_AD_, soluble solids, and acidity.

The variation on the *y*-axis (PC2) allowed for the differentiation between ‘Bergeron’ and ‘Goldrich’, explaining 28.6% of the variation. The main metabolites responsible for this separation were various organic acids and flavonoids, which played a significant role in each cultivar. In ‘Bergeron’, OA4, OA5, and F3 were prominent, along with G1 and G3 as glycosides and glucosylated compounds, and the amino acid AA4, which was particularly relevant, as well as the percentage of blush coverage. In ‘Goldrich’, the most significant organic acids and flavonoids were OA1, OA7, F1, and F4, along with fruit weight. Additionally, sucrose (C4) and certain amino acids (AA1, AA3, and AA6), which were relevant in ‘Goldrich’, following ‘Currot’, were also notable.

To provide a clear and detailed representation of the relationships between metabolites and cultivars, Figure 2 shows a heatmap of the metabolites identified in the ‘Currot’, ‘Goldrich’, and ‘Bergeron’ cultivars. Hierarchical clustering analysis was performed for both the metabolites and the samples of the different cultivars, allowing the observation of similarities and differences in metabolic profiles. The clustering of metabolites resulted in the formation of five clusters, visualized on the left side of Figure 2. From top to bottom, they are described as follows:

#### 2.5.1. First Cluster

This cluster comprises metabolites such as AA1, AA3, AA6, AC1, and G4, which show greater similarity between ‘Currot’ and ‘Goldrich’ cultivars, markedly differing from ‘Bergeron’. The presence of these metabolites is consistent in ‘Currot’ and ‘Goldrich’, whereas their abundance is significantly lower in ‘Bergeron’.

#### 2.5.2. Second Cluster

This cluster includes metabolites OA1, OA7, PA3, F1, F4, F5, C3, C4, and PD1, which have higher abundance in ‘Goldrich’, followed by ‘Bergeron’, and finally ‘Currot’. These metabolites, predominantly organic acids and flavonoids, are more prominent in ‘Goldrich’, indicating their importance in differentiating this cultivar.

#### 2.5.3. Third Cluster

Metabolites such as AA2, AA5, AK1, C1, C2, G2, G5, OA6, and PA2 are part of this cluster, showing high abundance in ‘Currot’, with lower levels in ‘Bergeron’ and ‘Goldrich’. This pattern suggests that these metabolites are characteristic of ‘Currot’ and contribute to its unique profile, highlighting the content of amino acids and carbohydrates like fructose and glucose.

#### 2.5.4. Fourth Cluster

This cluster includes metabolites OA3, OA5, PA1, F2, and F3, which present higher abundance in ‘Bergeron’, followed by ‘Goldrich’, and finally ‘Currot’. The presence of these metabolites is a key factor in the differentiation of ‘Bergeron’ from the other two cultivars. Like ‘Goldrich’, its differentiation is based on specific metabolites within the groups of organic acids and flavonoids.

#### 2.5.5. Fifth Cluster

Finally, the fifth cluster includes metabolites such as AA4, OA4, C5, G1, and G3, which show notable similarity between ‘Goldrich’ and ‘Currot’, differentiating from ‘Bergeron’, which showed the highest abundance.

These clusters allow for a detailed visualization of how metabolites are distributed and vary among the different cultivars, providing a clear understanding of the metabolic relationships that contribute to the differentiation of ‘Currot’, ‘Goldrich’, and ‘Bergeron’. The heatmap reveals that the ‘Currot’, ‘Goldrich’, and ‘Bergeron’ cultivars have distinctive metabolic profiles. The distribution and abundance of metabolites in the heatmap support the observations from the principal component analysis (PCA), where the main axes (PC1 and PC2) effectively differentiate the cultivars based on their metabolic and pomological profiles. The patterns observed in the heatmap provide a detailed visualization that complements the PCA, highlighting the key metabolites responsible for the variation and the relationships among the different cultivars.

## 3. Discussion

This section analyzes the metabolomic profile identified in the study, focusing on primary and secondary metabolites. Primary metabolites, such as amino acids, carbohydrates, and organic acids, are essential for metabolism and homeostasis [22]. Secondary metabolites, including flavonoids, glucosides, and glycosylated compounds, play key roles in plant defense and interactions [23]. The goal is to correlate metabolite profile changes with apricot fruit phenotyping data, linking analytical techniques to field phenotypic characterization.

In this section, we will focus on dissecting and analyzing the metabolomic profile observed in our study. Through the identification and quantification of a variety of metabolites, we have developed a deeper understanding of the underlying biochemical and physiological processes. The discussion will be structured around primary and secondary metabolites. Primary metabolites include amino acids, carbohydrates, and organic acids, all of which are fundamental for cellular metabolism and homeostasis [22]. On the other hand, secondary metabolites, such as flavonoids, glucosides, and glycosylated compounds, play crucial roles in plant defense, signaling, and interactions with the environment [23]. By evaluating these compounds, we aim to correlate the changes in compound profiles with the data obtained from phenotyping apricot fruits, thereby providing a comprehensive view of the relationship between the analytical techniques employed and the phenotypic characterization in the field.

### 3.1. Primary Metabolites

#### 3.1.1. Amino Acids

Amino acids in apricots are crucial components for the synthesis of proteins and enzymes, which are essential for a wide range of biological functions [24]. These amino acids are classified as primary metabolites due to their direct role in cell growth and development [25]. Among the amino acids, the most notable was asparagine, due to its high content at commercial maturity in the cultivars.

Asparagine plays a vital role in protein synthesis, contributing to the protein content and nutritional value of apricots. It also serves as a nitrogen reservoir, essential for its transport and storage during fruit growth and development [26]. Its levels in Prunus fruits vary depending on factors such as the cultivar, growth conditions, and ripening stage. Studies reveal that asparagine concentrations fluctuate during fruit development, aligning with nitrogen demand at various growth stages, and increase during ripening to support protein accumulation and other nitrogenous compounds [27,28].

This variability is also reflected in differences between cultivars. For instance, the ‘Currot’ cultivar exhibits a significantly higher concentration of asparagine (24.44 mg/g) compared to ‘Bergeron’ (8.93 mg/g) and ‘Goldrich’ (13.27 mg/g). The high concentration of asparagine in ‘Currot’ could influence organoleptic characteristics such as flavor, highlighting the importance of this amino acid in fruit quality, as it contributes to its sensory profile. This influence is particularly relevant in terms of quality and consumer acceptance.

#### 3.1.2. Carbohydrates

Apricots are rich in carbohydrates, mainly sugars, which significantly impact their nutritional value and sweet taste, improving acceptability and consumption [29]. Soluble solids, measured in °Brix, are a key quality indicator influenced by fructose, glucose, and sucrose, while myo-inositol (C3) and xylose (C5) remain at low concentrations (<3 mg/g) across all cultivars [30].

To better understand the relationship between carbohydrates and soluble solid content in various apricot cultivars, this study has revealed significant variations in the concentrations of these sugars among the ‘Goldrich’, ‘Bergeron’, and ‘Currot’ cultivars. On one hand, the sum of the quantified carbohydrates in each cultivar shows a correlation with the measured soluble solid content. ‘Bergeron’ had a total of 664 mg/g of carbohydrates, reflected in 14.3 degrees Brix, ‘Currot’ had 636 mg/g and 12.0 degrees Brix, and, finally, ‘Goldrich’ had 709 mg/g with 13.1 degrees Brix.

On the other hand, fructose and sucrose emerge as the main contributors to perceived sweetness, highlighting their importance in determining the organoleptic profile of apricots [31]. Fructose is the sweetest sugar among the three mentioned here. It has a higher sweetening power than glucose and sucrose, meaning that a smaller amount of fructose can produce a greater sensation of sweetness [32]. For instance, ‘Currot’ has the highest concentration of fructose (79.93 mg/g), which could significantly contribute to its perceived sweetness.

Although not as sweet as fructose, sucrose also significantly contributes to sweetness. It is a disaccharide composed of glucose and fructose, and its concentration is generally high in fruits perceived as sweeter [33]. In the case of ‘Goldrich’, which has the highest concentration of sucrose (546.98 mg/g), this high amount of sugar could be related to its level of sweetness.

It is noteworthy that, although it is less sweet than fructose and sucrose, glucose also contributes to the overall perception of sweetness. ‘Currot’ had the highest concentration of glucose (180.83 mg/g), which also influences its sweet flavor profile, making it superior to ‘Goldrich’ and ‘Bergeron’.

#### 3.1.3. Organic Acids

The organic acids present in apricots contribute to their characteristic acidic flavor [34]. According to phenotyping, acidity, expressed in grams of malic acid per 100 mL, is a key indicator of fruit quality. The balance between acidity and sugar determines the final taste of the fruit, which can be sweet if sugars predominate over acidity, or can counteract the sweetness of the sugars with a high level of acids [5,35].

The organic acids that most influence acidity are citric acid, malic acid, and quinic acid, while fumarate, succinate, and tartrate remained at low concentrations (<1.5 mg/g) across all cultivars, and formate was not detected in any. The correlation between organic acids and total acidity in the ‘Goldrich’, ‘Bergeron’, and ‘Currot’ cultivars is clear and directly proportional. The sum of the quantified organic acids in each cultivar is 306, 224, and 188 mg/g, respectively, corresponding to acidities of 2.64, 2.19, and 1.44 g/100 mL, respectivley.

Traditionally, apricot acidity is measured in grams of malic acid, due to its significant role in total acidity [36]. This study highlights that the malic acid/citric acid ratio varies by cultivar, as follows: ‘Bergeron’ has the highest malic acid content (109 mg/g), while ‘Currot’ shows a balanced ratio (~1/1), and, in ‘Goldrich’, citric acid predominates over malic acid (202 mg/g vs. 57 mg/g). Therefore, organic acids contribute differently to fruit acidity depending on the cultivar.

The high acidity content of ‘Goldrich’ suggests that, despite its high sucrose content, the acidity/sugar balance results in a less sweet taste compared to ‘Bergeron’. The latter has lower acidity and slightly less sugar content than ‘Goldrich’, but its balance suggests a sweeter taste. On the other hand, ‘Currot’, despite its lower sugar content, has higher concentrations of fructose and glucose along with low acidity, making it the ideal combination to achieve its characteristic sweetness.

Moreover, organic acids have natural preservative properties that help to prolong the fruit’s shelf life. Research has observed that ‘Goldrich’ and ‘Bergeron’ exhibited a longer shelf life compared to ‘Currot’, suggesting a potential relationship between acidity and fruit shelf life [18].

### 3.2. Secondary Metabolites

#### 3.2.1. Phenolic Acids

Ferulic acid, caffeic acid, and coumaric acid are three major phenolic acids identified in apricot fruits.

Ferulic acid is particularly noted for its ability to reduce oxidative stress and inflammation, as well as improve lipid profiles, which may lower the risk of cardiovascular diseases [37]. Additionally, it exhibits neuroprotective, antidiabetic, and anticancer potential, with applications in patented pharmaceuticals [38]. In the food industry, ferulic acid plays a role in preventing discoloration, acts as a precursor in the production of vanillin, and is used as a photoprotective agent in cosmetics [39]. Notably, the ‘Goldrich’ cultivar of apricots shows higher ferulic acid levels compared to ‘Bergeron’ and ‘Currot’, making it the most suitable option for dietary incorporation.

Meanwhile, caffeic acid is widely used to inhibit enzymatic browning in fruits and vegetables, while also stabilizing cosmetic formulations [40]. The lower levels of caffeic acid in the ‘Currot’ cultivar may suggest a faster browning process in comparison to ‘Goldrich’ and ‘Bergeron’.

On the other hand, coumaric acid has been suggested to offer antidiabetic benefits by modulating glucose metabolism [41]. Furthermore, similar to caffeic acid, coumaric acid shows promise in cosmetics, particularly for its anti-aging effects, as it helps protect the skin from oxidative damage [42]. In this regard, the ‘Currot’ cultivar exhibits significantly higher levels of coumaric acid, highlighting its value in health and cosmetic applications.

Overall, these compounds demonstrate remarkable antioxidant, anti-inflammatory, and antimicrobial properties, making them valuable for both health benefits and industrial applications [43]. Caffeic acid, meanwhile, has been extensively studied for its antioxidant, anti-inflammatory, and anticancer properties, although further clinical research is required to fully understand its mechanisms of action [44]. Coumaric acid, though less studied, also displays strong antioxidant and antimicrobial properties.

Together, these phenolic acids contribute not only to human health by mitigating oxidative stress and inflammation, but also hold important roles in food preservation and cosmetic industries. The distinct phenolic profiles across different apricot cultivars further emphasize their multifunctional potential in both health and industrial applications.

#### 3.2.2. Flavonoids

In apricots, the most important flavonoids related to color are anthocyanins [45]. Anthocyanins are responsible for the red, blue, and purple colors in the fruits.

In addition to their role in pigmentation, flavonoids also contribute to color stability, enhance the organoleptic qualities of apricots, and protect the fruit against oxidative stress [46]. Regarding their impact on human health, flavonoids improve overall health due to their antioxidant, anti-inflammatory, and cardioprotective properties, making this fruit not only nutritious, but also functional [47].

In our study, all identified flavonoids were classified as flavonols, which have been reported to exhibit low correlations with the skin color of the fruit [45]. However, they are closely associated with anthocyanins and other flavonoid subgroups through the phenylpropanoid pathway [48]. Despite being precursors of proanthocyanidins, their primary role is attributed to their antioxidant capacity. Although compounds such as catechin, myricitrin, quercetin, and rutin were not quantitatively analyzed, their presence was comparatively inferred through untargeted UPLC-QToF MS/MS analysis. This analysis revealed a general distribution pattern in which the ‘Goldrich’ cultivar exhibited higher levels of these flavonoids compared to ‘Bergeron’, with ‘Currot’ showing the lowest concentrations. These findings suggest significant implications for the nutritional and functional properties of apricots, underscoring the critical role of flavonoids in determining the overall quality of the fruit.

#### 3.2.3. Glycosides and Glucosylated Compounds

Glycosylated compounds consist of a sugar molecule attached to a non-sugar molecule, typically an organic compound. In apricots, these compounds significantly influence the fruit’s flavor, aroma, and overall quality [49]. The glycosides identified in the analyzed cultivars include kiwiionoside, neryl arabinofuranosyl-glucoside, vanillin glucose, zizybeoside I, and 3-hydroxy-beta-ionol 3-[glucosyl-(1->6)-glucoside]. These glycosides are notable for their contribution to the aromatic profile of apricots, as they can release volatile aromatic substances upon enzymatic hydrolysis [50]. Thus, the balance of these glycosides is essential to achieving the desired sensory profile, ensuring that the fruit is pleasant and enjoyable for consumers.

From the perspective of consumer health, glycosides offer several benefits. They possess antioxidant properties that neutralize free radicals, protecting cells from oxidative damage associated with aging and chronic diseases. Additionally, some glycosides exhibit anti-inflammatory and anticancer activities. Finally, they enhance digestive health by promoting the growth of beneficial bacteria in the intestine, improving digestion, and strengthening the immune system [51,52,53].

The significant differences found in the three cultivars regarding glycoside levels, analyzed through untargeted UPLC QToF MS/MS, suggest unique glycosylation patterns. Notably, zizybeoside I and 3-hydroxy-beta-ionol 3-[glucosyl-(1->6)-glucoside] were not detected in ‘Bergeron’ but were present in ‘Currot’ and ‘Goldrich’, reflecting distinct metabolic pathways. These cultivar-specific glycosylation mechanisms hold great potential as biomarkers for quality control in food safety [54].

## 4. Materials and Methods

### 4.1. Plant Material

The plant material used in this study comprises three apricot cultivars from the collection of the CEBAS-CSIC research center (Murcia, Spain). The cultivars analyzed were the French traditional cultivar ‘Bergeron’, the Spanish ‘Currot’, and the North American ‘Goldrich’ (Table 5). Quality parameters, corresponding to the pomological characteristics of the fruit, were evaluated, and the nutraceutical profile of the fruits was characterized. The plant material for these three cultivars was sourced from the experimental orchard owned by CEBAS-CSIC, located between the municipalities of Cieza and Calasparra in Murcia (southeastern Spain, 37° N latitude, 1° W longitude, 350 m altitude) in the year 2023.

### 4.2. Experimental Design and Testing

The fruit collection for the experiment was conducted on two trees of each cultivar, all of which were at least 10 years old and grown under the same, optimal conditions; therefore, the observed differences are mainly due to genetic background, as the fruits were harvested at a similar maturity stage under identical environmental conditions. In the first step, fifteen fruits from each cultivar were harvested when they reached their characteristic color (‘Currot’, yellow; ‘Bergeron’, light orange; and ‘Goldrich’, intense orange) and firmness close to 40 N to evaluate pomological characteristics such as fruit weight, chlorophyll index (I_AD_), skin color, blush color, flesh color, percentage of blush, and firmness. Then, these fruits were halved, generating two different mixes with the following purposes: The first sample mix, containing 15 fruit halves, was used to determine the soluble solid content and total acidity percentage, using three mixed replicates of 5 fruit halves each. Subsequently, the other mix with the remaining 15 halves was lyophilized, with three replicates considered for the application of analytical techniques such as ^1^H NMR and untargeted UPLC QToF MS/MS.

### 4.3. Pomological Traits Analysis

In this study, a range of pomological traits were assessed. First, fruit weight was recorded in grams using a Blauscal digital balance (model AH-600). The chlorophyll index was determined using a DA-meter (Sinteléia, Bologna, Italy), a portable Vis-NIR spectrometer that correlates fruit maturity with the chlorophyll absorbance difference index, known as I_AD_. Additionally, fruit color was evaluated with a Minolta colorimeter (CR-300; Minolta, Ramsey, NJ, USA). Three color measurements were taken on both the skin and flesh after calibration with a white porcelain reference plate. The CIELAB scale was used to determine the following three color coordinates: L*, a*, and b*. The hue angle (H° = arctangent(b*/a*)) was calculated [55], with values between 80 and 90 indicating a yellow coloration, values between 70 and 80 signifying an orange hue, and values below 70 representing a more reddish coloration. Firmness was measured using the TA.XT plus texture analyzer (South Hamilton, MA 01982, USA). The analysis involved compressing an area of 5 mm^2^, with the maximum force required for fruit deformation expressed in newtons (N) at a speed of 25 mm/min. The soluble solid content was measured using an Atago PAL-1 refractometer. Measurements were taken from crushed apricot flesh, with results expressed in °Brix. For acidity determination, 2 g of crushed sample was weighed and diluted in 30 mL of distilled water, using an automatic titration system (model 785 DMP Tinitro Metrohm Ltd., Switzerland). Finally, acidity was measured by neutralization with 0.1 N NaOH until reaching a pH of 8.1. The results were expressed in grams of malic acid per 100 mL.

### 4.4. Proton Nuclear Magnetic Resonance (^1^H NMR) Analysis

Three replicates of each lyophilized mix from the cultivars were prepared for analysis according to the protocol [56]. For this analysis, a Nuclear Magnetic Resonance (NMR) system coupled to a 500 MHz Bruker spectrometer (Bruker Biospin, Rheinstetten, Germany) equipped with a broadband 5 mm N2 CryoProbe Prodigy BBO was used. All samples were measured at 300.1 ± 0.1 K without rotation, with 4 test scans preceding the 32 experimental scans. The acquisition parameters were as follows: FID size = 64 K, spectral width = 12.4345 ppm, receiver gain = 28.5, acquisition time = 2.18 s, relaxation delay = 2 s, and line broadening = 0.50 Hz. Data acquisition was performed using the NOESY pre-saturation pulse sequence (Bruker 1D, noesypr1d) with water suppression via irradiation of the water frequency during the recycling and mixing times. Each spectrum underwent noise reduction based on multi-level signal deconvolution, followed by baseline correction and interpolation of signal areas. This process provides a “fingerprint” of the sample, offering an overview of the most represented metabolites at harvest time, with chemical shifts (δ) expressed in parts per million (ppm). The NMR equipment detects signals and records them as a frequency vs. intensity graph, known as the “acquisition spectrum”. The resulting ^1^H-NMR spectra were processed with the Chenomx NMR Suite version 8.3 (Chenomx, Edmonton, Canada) (Appendix A) to identify and quantify the metabolites of interest (Appendix A). All samples were calibrated using the signal from the internal standard (IS), deuterated trimethylsilylpropionic acid sodium salt (TSP-d4), and the pH was adjusted to around 6. The software includes a broad range of spectral data, enabling the detection of metabolites over 5–10 μM. Among the identified and/or quantified metabolites were the following: alanine, asparagine, isoleucine, phenylalanine, threonine, valine, choline, trigonelline, fructose, glucose, myo-Inositol, sucrose, xylose, epicatechin, citrate, formate, fumarate, malate, quinic acid, succinate, tartrate, and chlorogenate.

### 4.5. Untargeted UPLC QToF MS/MS Analysis

Three replicates of the lyophilized mix from each cultivar were weighed (50 mg each) and extracted using 1 mL of an 80/20 CH_3_OH/H_2_O (*v*/*v*) mixture, HPLC grade. The samples were mechanically shaken with a vortex, sonicated for three intervals of 30 s each, and then centrifuged for 10 min at 13,000× *g*, as previously described [57]. Additionally, the samples were mixed with glipizide (Sigma) at a concentration of 0.1 μg/mL, which served as an internal standard. Finally, the extracts were passed through a 13 mm PVDF syringe filter (0.22 μm, Millipore, Burlington, MD, USA) to ensure clarity and remove any particulate matter.

Ultra-Performance Liquid Chromatography coupled with Quadrupole Time-of-Flight mass spectrometry/mass spectrometry (UPLC-QToF-MS/MS) was conducted using a Waters ACQUITY UPLC I-Class System (Waters Corporation, Milford, MA, USA) paired with a Bruker Daltonics QToFMS mass spectrometer (maXis impact Series, resolution ≥ 55,000 FWHM, Bruker Daltonics, Bremen, Germany). Both positive [ESI(+)] and negative [ESI(−)] ionization modes were utilized. The UPLC separation employed an HSS T3 C18 column (100 × 2.1 mm, 1.8 μm particle size, Waters Corporation, Milford, MA, USA) at a flow rate of 0.3 mL/min. For the mobile phases, water with 0.01% formic acid (pH ~3.20) (PanReac AppliChem, Barcelona, Spain) was used as mobile phase (A), and acetonitrile with 0.01% formic acid (J. T. Baker, Phillipsburg, NJ, USA) was used as mobile phase (B). The gradient conditions used in this study are a slight modification of those employed in this assay [58]. The gradient started at 10% B and gradually increased to 90%. After maintaining this level for a few minutes, it rapidly decreased to 10% within 10 s and remained at this value until the end of the chromatogram. Nitrogen served as both the desolvation gas (flow rate of 8 L/min) and the nebulizing gas (pressure of 2.0 bar). The drying temperature was set to 200 °C, and the column temperature was maintained at 40 °C. The source voltage was set to 4.0 kV for ESI(−) and 4.5 kV for ESI(+). The MS analysis used High-Resolution QToF-MS with 24 eV for ESI(+) and 20 eV for ESI(−), utilizing broadband collision-induced dissociation (bbCID). MS data were recorded over an *m*/*z* range of 45–1200 Da. Calibration was performed externally before each sequence with a 10 mM sodium formate solution, delivered using a KNAUER Smartline Pump 100 equipped with a pressure sensor (KNAUER, Berlin, Germany). The calibration mixture consisted of 0.5 mL formic acid, 1.0 mL of 1.0 M sodium hydroxide, and an isopropanol/Milli-Q water solution (1:1, *v*/*v*).

### 4.6. Metabolite Extraction Protocol from Untargeted UPLC QToF MS/MS

The metabolite extraction process was conducted in several key stages. Initially, the sample extraction procedure was performed, followed by chromatographic analysis using UPLC QToF MS/MS in both positive and negative ionization modes. The results indicated that the positive ionization mode provided higher signal intensity. Firstly, a table was generated identifying 3210 peaks (Appendix A), with their corresponding retention times and m/z ratios, using the equipment’s software. These data were subsequently processed with the open-source software MetaboAnalyst to perform an ANOVA with a 95% confidence level, identifying 610 significantly different peaks. From these peaks, only those with retention times between 0.4 and 22 min in the chromatogram were selected, as this interval is effective for compound identification, excluding the dead time and the elution period with 100% non-polar mobile phase, resulting in 585 peaks. Subsequently, a minimum intensity threshold of 1000 units was applied, and matches were searched in various databases, such as KEGG (Kyoto Encyclopedia of Genes and Genomes) [59], HMDB (Human Metabolome Database) [60], and CEUMass. The m/z ratios and their corresponding tandem mass spectrometry (MS/MS) fragmentation spectra were compared, tentatively identifying the following 12 compounds (Figure 3): caffeic acid, coumaric acid, ferulic acid, catechin, myricitrin, quercetin, rutin, kiwiionoside, neryl arabinofuranosyl-glucoside, vanilloyl glucose, zizybeoside I, and 3-hydroxy-beta-ionol 3-[glucosyl-(1->6)-glucoside]. The accuracy of the measurements was evaluated using parameters such as error and mSigma, providing high confidence in the tentative identifications [61,62]. In mass spectrometry, acceptable error values should be within ±5 ppm. mSigma is a statistical measure that quantifies the normalized standard deviation of the differences between theoretical and measured masses, considering multiple isotopic peaks of the ion in question. To ensure reliable and precise compound identification, an mSigma value below 10 is preferred, while values above 30 should be considered with caution and may require additional verification.

### 4.7. Data Analysis

Data from the analyses are presented as mean ± standard deviation. One-way ANOVA, followed by Tukey’s test at a 5% significance level, was performed to identify significant differences using INFOSTAT v18 software (Universidad Nacional de Córdoba, Argentina). For metabolite compound data analysis, a principal component analysis (PCA) and heatmap visualizations were performed using the free software R (R version 4.3.2, RStudio team). PCA biplots were created using the “factoextra”, “FactoMineR”, “readxl”, and “textshape” packages, while the “pheatmap” package was used for heatmap visualizations.

## 5. Conclusions

This research provides a detailed characterization of the metabolomic profile of three apricot cultivars (‘Bergeron’, ‘Currot’, and ‘Goldrich’), highlighting significant variations in their biochemical composition. These results shed light on the possibility of measuring these compounds in populations derived from these parents to identify marker–trait associations of interest.

Through the application of ^1^H NMR and untargeted UPLC-QToF MS/MS, a wide range of primary and secondary metabolites—including amino acids, carbohydrates, organic acids, flavonoids, and glycosylated compounds—were identified and quantified. These metabolites play a crucial role in determining the sensory attributes of the fruit, its nutritional value, and its potential health benefits. Among the studied cultivars, ‘Bergeron’ and ‘Goldrich’ exhibited higher concentrations of organic acids, flavonoids such as epicatechin, and sucrose. These compounds contribute to their distinctive acidity-to-sugar balance. In contrast, ‘Currot’ showed elevated levels of amino acids, particularly asparagine, along with higher concentrations of sugars, notably fructose and glucose, which enhance its characteristic sweetness.

The combined use of heatmaps and PCA provides a robust methodology for analyzing and visualizing the variability and relationships within our metabolite data, offering a comprehensive understanding of the factors that differentiate these apricot cultivars. This approach not only deepens our insight into the chemical composition underlying fruit quality but also holds practical applications for breeding programs. By selecting cultivars with optimal metabolomic profiles—such as those rich in key organic acids or sugars—breeders can develop apricots with improved organoleptic properties, higher nutritional value, and enhanced health benefits. For instance, cultivars with elevated levels of antioxidant compounds like flavonoids (e.g., epicatechin) could meet the growing consumer demand for high-quality fruits. The results of this study lay the groundwork for future research exploring the genetic and environmental factors influencing apricot metabolism. Expanding the analysis to a broader range of cultivars and different locations will provide deeper insights into the biochemical diversity of apricots, ultimately contributing to the sustainable production of superior cultivars. Moreover, this approach enables the study of metabolomic evolution throughout the maturation process and even in cultivars subjected to different stress conditions for further analysis.

## Figures and Tables

**Figure 1 plants-14-01000-f001:**
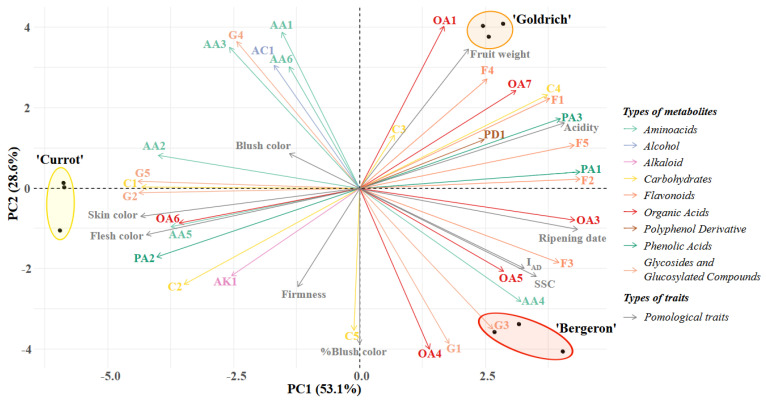
Principal component analysis (PCA) of metabolites identified by both analytical techniques (^1^H NMR and untargeted HPLC-MS) and pomological traits in each of the studied cultivars.

**Figure 2 plants-14-01000-f002:**
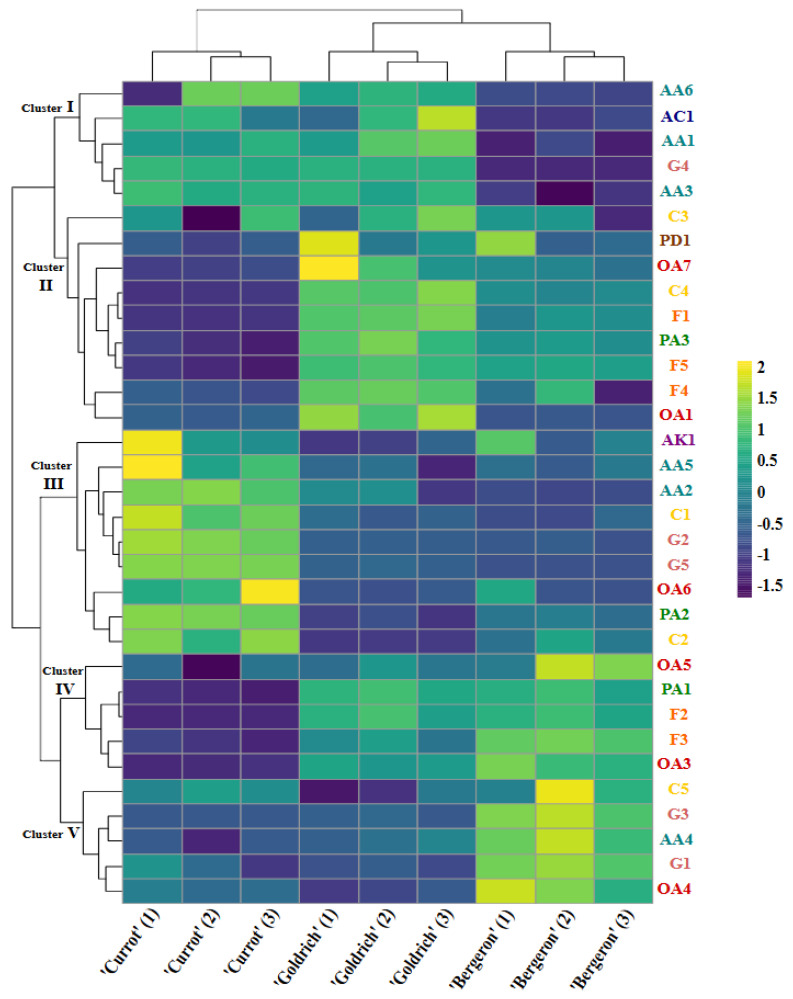
Heatmap representation of metabolites identified by both analytical techniques (^1^H NMR and untargeted HPLC-MS) in each replicate of each cultivar.

**Figure 3 plants-14-01000-f003:**
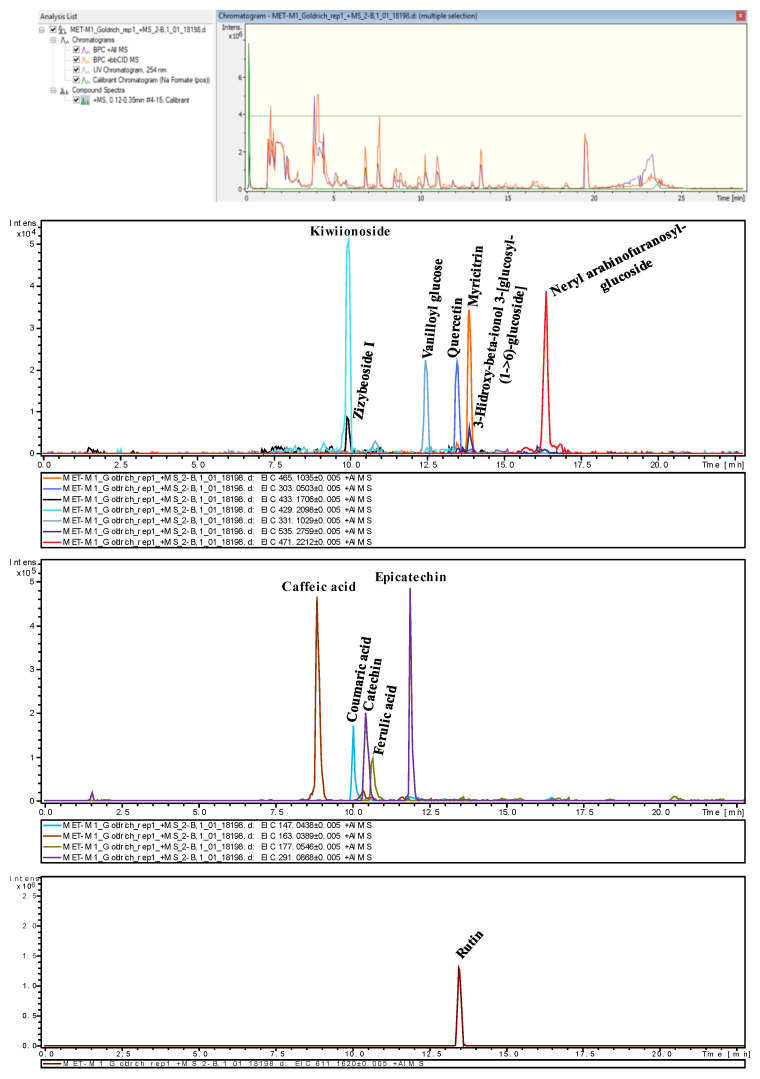
Representative example of tentatively identified metabolite peaks in untargeted HPLC-MS chromatograms via an Extracted Ion Chromatogram in one of the cultivars (‘Goldrich’).

**Table 1 plants-14-01000-t001:** Overview of descriptive statistics for pomological traits of fruit assessed at harvest among three apricot cultivars.

Apricot Cultivar	Trait	Mean ± SD	Apricot Cultivar		Trait		Mean ± SD
‘Bergeron’	Ripening date	167	‘Goldrich’		Ripening date		157
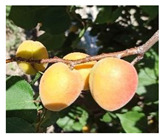	Fruit weight	40.41 ± 4.45	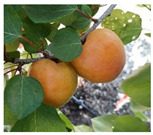		Fruit weight		86.57 ± 14.39
	I_AD_	0.34 ± 0.13			I_AD_		0.22 ± 0.08
	Skin color	77.82 ± 2.51			Skin color		72.61 ± 1.21
	Blush color	48.11 ± 8.11			Blush color		56.09 ± 6.23
	% Blush color	26.25 ± 6.78			% Blush color		9.28 ± 5.35
	Flesh color	73.83 ± 1.84			Flesh color		70.70 ± 0.71
	Firmness	47.22 ± 12.74			Firmness		34.44 ± 15.63
	SSC	14.30 ± 0.53			SSC		13.13 ± 0.47
	Acidity	2.19 ± 0.11			Acidity		2.64 ± 0.09
	**Apricot cultivar**		**Trait**	**Mean ± SD**			
	‘Currot’		Ripening date	130			
	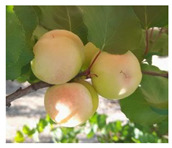		Fruit weight	38.84 ± 3.32			
			I_AD_	0.10 ± 0.05			
			Skin color	98.34 ± 2.71			
			Blush color	58.39 ± 12.92			
			% Blush color	18.75 ± 7.42			
			Flesh color	98.14 ± 2.18			
			Firmness	49.63 ± 10.11			
			SSC	12.03 ± 0.75			
			Acidity	1.44 ± 0.05			

**Table 2 plants-14-01000-t002:** Classification and quantification of metabolites identified by ^1^H NMR in three apricot cultivars.

Class	ID	Compound	Formula	^1^H (ppm) ^a^	Multiplicity ^b^	‘Bergeron’ (mg/g) ^cd^	‘Currot’ (mg/g) ^cd^	‘Goldrich’ (mg/g) ^cd^
Amino acids	AA1	Alanine	C_3_H_7_NO_2_	1.49	d	0.47 ± 0.06 b	0.80 ± 0.04 a	0.88 ± 0.09 a
	AA2	Asparagine	C_4_H_8_N_2_O_3_	2.95	dd	8.93 ± 0.32 b	24.44 ± 1.43 a	13.27 ± 5.25 b
	AA3	Isoleucine	C_6_H_13_NO_2_	0.98	t	0.11 ± 0.01 b	0.18 ± 0.01 a	0.17 ± 0.01 a
	AA4	Phenylalanine	C_6_H_11_NO_2_	7.40	m	0.21 ± 0.02 a	0.12 ± 0.02 b	0.15 ± 0.01 b
	AA5	Threonine	C_4_H_9_NO_3_	1.32	d	0.22 ± 0.01 b	0.30 ± 0.05 a	0.20 ± 0.03 b
	AA6	Valine	C_5_H_11_NO_2_	1.00	d	0.163 ± 0.003 a	0.23 ± 0.07 a	0.24 ± 0.01 a
Alcohol	AC1	Choline	(C_5_H_14_NO)^+^	3.20	s	0.032 ± 0.002 a	0.06 ± 0.01 a	0.06 ± 0.02 a
Alkaloid	AK1	Trigonelline	C_7_H_7_NO_2_	9.10	s	0.04 ± 0.01 a	0.04 ± 0.01 a	0.029 ± 0.003 a
Carbohydrates	C1	Fructose	C_6_H_12_O_6_	4.11	m	54.26 ± 3.08 b	79.93 ± 4.48 a	56.72 ± 1.86 b
	C2	Glucose	C_6_H_12_O_6_	5.22	d	139.68 ± 15.47 b	180.83 ± 14.24 a	100.75 ± 1.00 c
	C3	Myo-Inositol	C_6_H_12_O_6_	3.30	t	2.57 ± 0.20 a	2.58 ± 0.30 a	2.74 ± 0.20 a
	C4	Sucrose	C_12_H_22_O_11_	5.40	d	464.65 ± 4.89 b	370.64 ± 3.66 c	546.98 ± 14.85 a
	C5	Xylose	C_5_H_10_O_5_	5.10	d	2.46 ± 0.34 a	2.22 ± 0.07 ab	1.85 ± 0.22 b
Flavonoid	F1	Epicatechin	C_15_H_14_O_6_	6.05	d	0.44 ± 0.06 b	0.03 ± 0.01 c	0.79 ± 0.05 a
Organic Acids	OA1	Citrate	(C_6_H_5_O_7_)^−3^	2.74	dd	64.12 ± 1.86 b	73.98 ± 4.79 b	202.27 ± 22.37 a
	OA2	Formate	(CHO_2_)^−^	8.45	s	Not detected	Not detected	Not detected
	OA3	Fumarate	(C_4_H_2_O_4_)^−2^	6.53	s	0.025 ± 0.002 a	0.0126 ± 0.0003 c	0.0218 ± 0.0007 b
	OA4	Malate	(C_4_H_4_O_5_)^−2^	2.39	dd	109.16 ± 13.25 a	71.34 ± 4.00 b	57.56 ± 5.22 b
	OA5	Quinate	(C_7_H_11_O_6_)^−^	1.95	dd	49.33 ± 3.95 a	42.32 ± 3.04 a	45.03 ± 1.30 a
	OA6	Succinate	(C_4_H_4_O_4_)^−2^	2.49	s	0.04 ± 0.03 ab	0.09 ± 0.03 a	0.025 ± 0.003 b
	OA7	Tartrate	(C_4_H_4_O_6_)^−2^	7.10	s	0.91 ± 0.05 ab	0.66 ± 0.03 b	1.23 ± 0.26 a
Polyphenol Derivative	PD1	Chlorogenate	(C_16_H_17_O_9_)^−^	7.63	d	1.64 ± 0.87 a	0.95 ± 0.16 a	± 0.84 a

^a 1^H (ppm): Proton chemical shift in parts per million. ^b^ Multiplicity: Indicates the splitting pattern of the NMR signal (s = singlet, d = doublet, t = triplet, m = multiplet, dd = doublet of doublets). ^c^ ‘Bergeron’, ‘Currot’, and ‘Goldrich’: Refer to the concentrations in mg per g of fruit in respective samples. ^d^ Different letters indicate significant differences between the concentrations (mg/g) for each cultivar, according to the Tukey test (*p*-value < 0.05).

**Table 3 plants-14-01000-t003:** Classification of tentatively identified metabolites in three apricot cultivars using untargeted HPLC-MS assisted by databases.

Class	ID	Tentative Compound	Formula	Theoretical Mass (*m*/*z*)	Measured Mass (*m*/*z*)	RT (min)	Error	mSigma	Adduct	Database
Phenolic Acids	PA1	Caffeic acid	C_9_H_8_O_4_	163.0390	163.0389	9.03	0.2	8.7	[M−H_2_O+H]^+^	KEGG
	PA2	Coumaric acid	C_9_H_8_O_3_	147.0441	147.0438	10.06	2.0	6.8	[M−H_2_O+H]^+^	KEGG
	PA3	Ferulic acid	C_10_H_10_O_4_	177.0546	177.0546	10.76	0.0	2.9	[M−H_2_O+H]^+^	KEGG
Flavonoids	F2	Catechin	C_15_H_14_O_6_	291.0863	291.0868	10.43	−1.8	6.2	[M+H]^+^	HMDB
	F3	Myricitrin	C_21_H_20_O_12_	465.1028	465.1035	13.92	−1.6	1.2	[M+H]^+^	HMDB
	F4	Quercetin	C_15_H_10_O_7_	303.0499	303.0503	13.50	−1.1	13.9	[M+H]^+^	HMDB
	F5	Rutin	C_27_H_30_O_16_	611.1607	611.1620	13.54	−2.2	16.0	[M+H]^+^	HMDB
Glycosides and Glucosylated Compounds	G1	Kiwiionoside	C_19_H_34_O_9_	429.2095	429.2098	10.00	−0.8	5.3	[M+Na]^+^	HMDB
	G2	Neryl arabinofuranosyl-glucoside	C_21_H_36_O_10_	471.2201	471.2212	16.37	−2.4	6.4	[M+Na]^+^	HMDB
	G3	Vanilloyl glucose	C_14_H_18_O_9_	331.1024	331.1029	12.54	−1.5	3.0	[M+H]^+^	HMDB
	G4	Zizybeoside I	C_19_H_28_O_11_	433.1704	433.1706	9.97	−0.3	29.9	[M+H]^+^	HMDB
	G5	3-Hydroxy-beta-ionol 3-[glucosyl-(1->6)-glucoside]	C_25_H_42_O_12_	535.2749	535.2759	13.94	−1.9	11.2	[M+H]^+^	HMDB

**Table 4 plants-14-01000-t004:** Statistical comparison of relative area units of tentatively identified metabolites in three apricot cultivars using untargeted HPLC-MS assisted by databases.

Class	ID	Tentative Compound	‘Bergeron’ (ra)	‘Currot’ (ra)	‘Goldrich’ (ra)
Phenolic Acids	PA1	Caffeic acid	4,609,160 ± 463,508 a	532,861 ± 219,791 b	4,781,477 ± 368,002 a
	PA2	Coumaric acid	1,040,945 ± 90,857 b	2,140,665 ± 76,633 a	508,567 ± 119,475 c
	PA3	Ferulic acid	917,099 ± 21,049 b	640,206 ± 38,056 c	1,067,681 ± 50,773 a
Flavonoids	F2	Catechin	2,424,284 ± 249,228 a	17,918 ± 2567 b	2,400,544 ± 330,572 a
	F3	Myricitrin	362,724 ± 12,724 a	138,008 ± 19,084 c	256,125 ± 32,223 b
	F4	Quercetin	94,999 ± 67,109 ab	67,531 ± 9353 b	180,915 ± 5506 a
	F5	Rutin	9,012,800 ± 178,088 b	4,324,298 ± 458,177 c	10,174,290 ± 289,308 a
Glycosides and Glucosylated Compounds	G1	Kiwiionoside	833,782 ± 31,998 a	591,256 ± 96,070 b	544,706 ± 19,783 b
	G2	Neryl arabinofuranosyl-glucoside	293,059 ± 44,975 b	1,559,538 ± 104,179 a	364,621 ± 18,820 b
	G3	Vanilloyl glucose	1,100,944 ± 183,073 a	46,613 ± 5633 b	105,084 ± 60,922 b
	G4	Zizybeoside I	Not detected b	77,605 ± 4379 a	77,824 ± 301 a
	G5	3-Hydroxy-beta-ionol 3-[glucosyl-(1->6)-glucoside]	Not detected c	128,088 ± 2514 a	14,932 ± 3828 b

Different letters indicate significant differences between the arbitrary units (au) for each cultivar, according to the Tukey test (*p*-value < 0.05).

**Table 5 plants-14-01000-t005:** Description of three apricot cultivars with interest in plant breeding improvement.

Cultivar	Pedigree	Self-Compatibility	Sharka Resistance (PPV-D)	Flowering	Ripening	Skin Color
‘Bergeron’	Unknown	Self-compatible	No	Late	Late	Light Orange
‘Currot’	Unknown	Self-compatible	No	Early	Early	Light Yellow/Pink
‘Goldrich’	‘Sunglo × Perfection’	Self-incompatible	Yes	Late	Late	Orange

## Data Availability

The data that support the findings of this study are available from the corresponding author upon reasonable request.

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
