# Peer review of "Nutraceutical Profile Characterization in Apricot (*Prunus armeniaca* L.) Fruits"

_plants, 2025, doi:10.3390/plants14071000_

Round 1
Reviewer 1 Report
Comments and Suggestions for Authors
The manuscript titled “ Nutraceutical profile characterization in apricot (Prunus armeniaca L.) fruits” appears to be well-executed and written. The authors employ modern techniques such as MS and NMR, which enhance the study's analytical rigor. However, the primary challenge lies in the relatively weak novelty of the research. Below are specific points for improvement:
- Novelty and emphasis:
Apricots have already been extensively studied from a phytochemical perspective. To strengthen the manuscript, the authors should emphasize the unique aspects of their research more clearly. For instance, highlighting any novel findings or methodologies not previously reported in the literature would add value. - Standards and reagents:
The manuscript lacks information on the purity and source of the standards and reagents used in the study. Providing these details is essential for ensuring reproducibility and transparency. - Lipophilic metabolites:
While the study focuses on certain phytochemicals, it overlooks lipophilic secondary metabolites such as tocopherols and carotenoids, which are known to be present in apricots. Including a discussion on these compounds in this study would provide a more comprehensive understanding of apricot phytochemistry. - Phytochemical changes during ripening:
The manuscript would benefit from a discussion on how phytochemical composition changes during fruit ripening. This aspect is crucial for understanding the dynamics of bioactive compounds in apricots over time. - Reducing redundancy in discussion:
The discussion sections on amino acids, carbohydrates, and organic acids could be condensed, as these topics are well-documented for apricots. Instead, more focus should be placed on less-explored areas to avoid redundancy. - Antioxidant activity:
Rather than repeatedly mentioning antioxidant activity for each secondary metabolite, it would be more effective to include a concise paragraph summarizing this topic comprehensively. - Conclusions:
The conclusions section could be streamlined by omitting specific details such as exact concentrations or unnecessary parenthetical information (e.g., “malic and citric acid”). Additionally, the novelty aspect should be explicitly addressed in this section to reinforce the study's contribution to the field.
Author Response
Reviewer 1
The manuscript titled “Nutraceutical profile characterization in apricot (Prunus armeniaca L.) fruits” appears to be well-executed and written. The authors employ modern techniques such as MS and NMR, which enhance the study's analytical rigor. However, the primary challenge lies in the relatively weak novelty of the research. Below are specific points for improvement:
1. Novelty and emphasis:
Apricots have already been extensively studied from a phytochemical perspective. To strengthen the manuscript, the authors should emphasize the unique aspects of their research more clearly. For instance, highlighting any novel findings or methodologies not previously reported in the literature would add value.
Authors:
I agree with your valuable comment. However, although apricots have been extensively studied at the phytochemical level, this manuscript offers a new perspective. There are very few studies that quantify metabolites in apricots using ¹H Nuclear Magnetic Resonance (NMR), and even fewer that combine this approach with untargeted Ultra-Performance Liquid Chromatography coupled with Quadrupole Time-of-Flight Mass Spectrometry/Mass Spectrometry (UPLC-QToF-MS/MS) for polyphenolic compounds. This idea has been incorporated into the revised version of the manuscript (lines: 80-85).
2. Standards and reagents:
The manuscript lacks information on the purity and source of the standards and reagents used in the study. Providing these details is essential for ensuring reproducibility and transparency.
Authors:
Thank you very much for your suggestion. Proton Nuclear Magnetic Resonance (H NMR) analysis has been utilized to identify and quantify metabolites such as alanine, asparagine, isoleucine, phenylalanine, threonine, valine, choline, trigonelline, fructose, glucose, myo-inositol, sucrose, xylose, epicatechin, citrate, formate, fumarate, malate, quinic acid, succinate, tartrate, and chlorogenate. As explained in lines 588–592, all samples were calibrated using the signal from the internal standard, deuterated trimethylsilylpropionic acid sodium salt (TSP-d4), and the pH was adjusted to approximately 6. Additionally, the NMR equipment detects signals and records them as a frequency versus intensity graph, known as the acquisition spectrum.
To provide greater clarity and understanding, we have included new supplementary material (Figure S1), which contains the H NMR spectra of the three cultivars (‘Bergeron,’ ‘Currot,’ and ‘Goldrich’) analyzed using the Chenomx NMR Suite, version 8.3. In the second method, untargeted UPLC-QToF-MS/MS analysis, sample preparation (lines 598–604) is described in detail, along with the specifications and conditions of the equipment, column, mobile phases, gradients, etc. If you require additional information to further ensure the reproducibility of the assay, please let us know what you would like to have included.
3. Lipophilic metabolites:
While the study focuses on certain phytochemicals, it overlooks lipophilic secondary metabolites such as tocopherols and carotenoids, which are known to be present in apricots. Including a discussion on these compounds in this study would provide a more comprehensive understanding of apricot phytochemistry.
Authors:
Thank you for your valuable comment. I fully understand the importance of these secondary metabolites, as they are among the most significant compounds in apricots. However, the objective of this manuscript was to focus on other compounds associated with secondary metabolism, such as phenolic acids, flavonoids, glycosides, and glucosylated compounds, in addition to those related to primary metabolism. In the case of apricots, carotenoids and polyphenols are two groups of secondary metabolites that have been extensively studied due to their nutritional and functional relevance. Nonetheless, carotenoids seem to have been explored more thoroughly. For this reason, we decided to emphasize other secondary metabolites, as mentioned earlier. Like the former, these compounds also play a crucial role in human health, including their antioxidant and anti-inflammatory properties.
4. Phytochemical changes during ripening:
The manuscript would benefit from a discussion on how phytochemical composition changes during fruit ripening. This aspect is crucial for understanding the dynamics of bioactive compounds in apricots over time.
Authors:
Thank you for your comment. I truly appreciate it, as you provide valuable ideas for further analysis. In our case, the main objective was to investigate differences among parents rather than among different ripening stages. The purpose of focusing on differences among parents was to explore the possibility of analyzing these compounds in segregating populations. This approach could enable us to identify marker-trait associations after genotyping, allowing us to link specific loci to important metabolic compounds (lines: 97-100). In our study, we collected fruits under similar conditions prior to reaching consumer maturity.
5. Reducing redundancy in discussion:
The discussion sections on amino acids, carbohydrates, and organic acids could be condensed, as these topics are well-documented for apricots. Instead, more focus should be placed on less-explored areas to avoid redundancy.
Authors:
I agree; this discussion section has been condensed to reduce redundancy, reducing at least 14 lines from the text as follows: 3. Discussion (lines 353–359), 3.1.1. Amino Acids (lines 379–385), 3.1.2. Carbohydrates (lines 394–398), and 3.1.3. Organic Acids (lines 434–438).
6. Antioxidant activity:
Rather than repeatedly mentioning antioxidant activity for each secondary metabolite, it would be more effective to include a concise paragraph summarizing this topic comprehensively.
Authors:
Thank you very much for your time and valuable comment. In this section I have modified and restructured the paragraphs in order to avoid to repeat the word antioxidant activity again and again, especially in the phenolic acid section.
7. Conclusions:
The conclusions section could be streamlined by omitting specific details such as exact concentrations or unnecessary parenthetical information (e.g., “malic and citric acid”). Additionally, the novelty aspect should be explicitly addressed in this section to reinforce the study's contribution to the field.
Authors:
I agree with your suggestion, In the new version of the conclusion section I have omitted unnecessary information, and I have reinforced the idea of the potential contribution of this work.
Thank you very much for your suggestions and comments; they have truly helped improve the manuscript. All these modifications are reflected in the new version of the manuscript in green.
Reviewer 2 Report
Comments and Suggestions for Authors
The ms entitled 'Nutraceutical profile characterization in apricot (Prunus armeniaca L.) fruits' (authors: Germán Ortuño-Hernández, Marta Silva, Rosa Toledo, Helena Ramos, Ana Reis-Mendes, David Ruiz Pedro Martínez-Gómez, Isabel M.P.L.V.O. Ferreira and Juan Alfonso Salazar) aims to characterize the metabolomic profiles of three reference apricot cultivars (‘Bergeron,’ ‘Currot,’ and ‘Goldrich’) using ¹H NMR spectroscopy and untargeted UPLC-QToF MS/MS to support plant breeding by correlating metabolomic data with fruit phenotyping. The primary objective was to identify and quantify key metabolites influencing fruit quality from a nutraceutical perspective.
The ms undoubtedly reports an huge amount of very interesting analytical data but, due to methodological shortcomings, it can be considered as merely descriptive of three populations of fruits (15) of the three cultivars collected at a single moment of their ripening curve without having identified unique reference parameters relating to the physiological conditions of the trees and the actual ripening stage of the fruits.
Since the secondary metabolites analyzed are strictly linked to the plant's response to environmental stress and to the fruit ripening curve, the interannual variation that can occur as a function of the physiological state of the plant and that which can be found during the fruit ripening curve can be as large as the variability existing between one cultivar and another.
In particular, it is worth noting the strong variability detected by the authors in the flesh firmess of the three cultivars, demonstrating that the sample of fruits analyzed presented very different stages of ripening. In my opinion, it would have been necessary to clearly identify, in a unique and repeatable way over the years, the stage of fruit ripening at which to carry out the analyses.
Similarly, also from Figure 2 of the Heatmap representation of metabolites a clear difference can be seen between the three samples of fruit analysed.
In my opinion, it would have been much more interesting to verify the metabolic profile of the three cultivars in different conditions of water stress to hypothesize the best cultivation technique that determines the greatest accumulation of metabolites and the best organoleptic quality of the fruits, without which the commercial value of the same would be scarce, making the research itself lose its meaning.
However, the ms contains extremely interesting data that should be adequately valorised probably with a more careful repetition of the analyses in different years.
Finally, it should be underlined that the keywords referred to are partly already contained in the title and therefore redundant.
Author Response
Reviewer 2
Comments and Suggestions for Authors
The ms entitled 'Nutraceutical profile characterization in apricot (Prunus armeniaca L.) fruits' (authors: Germán Ortuño-Hernández, Marta Silva, Rosa Toledo, Helena Ramos, Ana Reis-Mendes, David Ruiz Pedro Martínez-Gómez, Isabel M.P.L.V.O. Ferreira and Juan Alfonso Salazar) aims to characterize the metabolomic profiles of three reference apricot cultivars (‘Bergeron,’ ‘Currot,’ and ‘Goldrich’) using ¹H NMR spectroscopy and untargeted UPLC-QToF MS/MS to support plant breeding by correlating metabolomic data with fruit phenotyping. The primary objective was to identify and quantify key metabolites influencing fruit quality from a nutraceutical perspective.
The ms undoubtedly reports an huge amount of very interesting analytical data but, due to methodological shortcomings, it can be considered as merely descriptive of three populations of fruits (15) of the three cultivars collected at a single moment of their ripening curve without having identified unique reference parameters relating to the physiological conditions of the trees and the actual ripening stage of the fruits.
Authors:
I really appreciate your valuable comments. The fruits were harvested at their optimal moment when they reached their characteristic color and a firmness close to 40N, as it is sometimes difficult to establish a single criterion. Moreover, it should be noted that the characteristic color of each variety can often influence the firmness of the fruits. This is why it is so challenging to have a single criterion to determine the ripening date.
Since the secondary metabolites analyzed are strictly linked to the plant's response to environmental stress and to the fruit ripening curve, the interannual variation that can occur as a function of the physiological state of the plant and that which can be found during the fruit ripening curve can be as large as the variability existing between one cultivar and another.
Authors:
I understand your concern; however, the fruits were harvested according to the optimal conditions known for each of them, based on their maximum color potential and with adequate firmness to ensure the minimum quality standards (commercial maturity). Regarding environmental conditions, I understand that it would be interesting to measure over different years to confirm similar differences in these compounds. However, we can say that the evaluated varieties were located in the same orchard and, therefore, responded to the same environmental conditions (lines: 540-541).
In particular, it is worth noting the strong variability detected by the authors in the flesh firmess of the three cultivars, demonstrating that the sample of fruits analyzed presented very different stages of ripening. In my opinion, it would have been necessary to clearly identify, in a unique and repeatable way over the years, the stage of fruit ripening at which to carry out the analyses.
Authors:
As mentioned above, establishing a single harvest criterion was challenging due to the significant differences among the cultivars, partly attributable to their distinct genetic backgrounds. Consequently, we adopted a dual criterion that considered skin color (‘Currot’: yellow with hº values above 85; ‘Bergeron’: light orange with hº values between 75 and 85; and ‘Goldrich’: intense orange with hº values below 75) alongside firmness of approximately 40N (lines: 543-544). Firmness values close to or above 40N are generally regarded as suitable for commercial harvest.
Similarly, also from Figure 2 of the Heatmap representation of metabolites a clear difference can be seen between the three samples of fruit analysed.
In my opinion, it would have been much more interesting to verify the metabolic profile of the three cultivars in different conditions of water stress to hypothesize the best cultivation technique that determines the greatest accumulation of metabolites and the best organoleptic quality of the fruits, without which the commercial value of the same would be scarce, making the research itself lose its meaning.
Authors:
This perspective may be interesting for future studies considering different stress conditions. However, in this case, the objective was to identify differences among parents at a specific ripening stage (commercial maturity) and even assess whether segregation of these traits could exist in populations derived from these parents. This would allow the analysis of these traits in segregating populations that have been previously genotyped to search for marker-trait associations.
However, the ms contains extremely interesting data that should be adequately valorised probably with a more careful repetition of the analyses in different years.
Finally, it should be underlined that the keywords referred to are partly already contained in the title and therefore redundant.
Authors:
I agree; I have replaced 'apricot' and Prunus armeniaca with 'stone fruits' and 'nutraceutical profile' with 'nutribreeding'."
Authors:
Thank you very much for your valuable suggestions and comments. They have been instrumental in improving the manuscript. All the changes have been carefully implemented and are highlighted in green in the revised version for easy reference. Your input is greatly appreciated.
Round 2
Reviewer 2 Report
Comments and Suggestions for Authors
The review of the ms carried out by the Authors appears convincing and exhaustive, despite the methodological limitations found, which are however very difficult to resolve in a very heterogeneous system such as an apricot orchard.
The characteristics of the fruits (qualitative and nutraceutical) are influenced by multiple factors, both environmental and cultural, which is why it is always difficult to be able to standardize the stage of ripening of the sampled fruits at harvest.
The work is very interesting but I would suggest the Authors to take better care of the fruit sampling phase for a next step to minimize environmental influences on the sample of fruits chosen.
The environmental characteristics of the fruits (level of incident radiation) and physiological characteristics of the plants (water potential, etc.) should be carefully evaluated and described.
The ms is interesting and can be further developed in a second test phase.